# $G$olden $G$oose 🦢: A Simple Trick to Synthesize Unlimited RLVR Tasks from Unverifiable Internet Text

Ximing Lu [1 2]  David Acuna [1]  Jaehun Jung [1]  Jian Hu [1]  Di Zhang [1]  Shizhe Diao [1]  Yunheng Zou [1]
Shaokun Zhang [1]  Brandon Cui [1]  Mingjie Liu [1]  Hyunwoo Kim [1]  Prithviraj Ammanabrolu [1 3]  Jan Kautz [1]
Yi Dong [* 1]  Yejin Choi [* 1]

## Abstract

Reinforcement Learning with Verifiable Rewards (RLVR) has become a cornerstone for unlocking complex reasoning in Large Language Models (LLMs). Yet, scaling up RL is bottlenecked by limited existing verifiable data, where improvements increasingly saturate over prolonged training. To overcome this, we propose $G$**olden** $G$**oose**, a simple trick to synthesize unlimited RLVR tasks from unverifiable internet text by constructing a multiple-choice question-answering version of the fill-in-the-middle task. Given a source text, we prompt an LLM to identify and mask key reasoning steps, then generate a set of diverse, plausible distractors. This enables us to leverage reasoning-rich unverifiable corpora typically excluded from prior RLVR data construction (e.g., science textbooks) to synthesize $G$**ooseReason-0.7M**, a large-scale RLVR dataset with over 0.7 million tasks spanning mathematics, programming, and general scientific domains. Empirically, *GooseReason* effectively revives models saturated on existing RLVR data, yielding robust, sustained gains under continuous RL and achieving new state-of-the-art results for 1.5B and 4B-Instruct models across 15 diverse benchmarks. Finally, we deploy *Golden Goose* in a real-world setting, synthesizing RLVR tasks from raw FineWeb scrapes for the cybersecurity domain, where no prior RLVR data exists. Training Qwen3-4B-Instruct on the resulting data $G$**ooseReason-Cyber** sets a new state-of-the-art in cybersecurity, surpassing a 7B domain-specialized model with extensive domain-specific pre-training and post-training. This highlights the potential of automatically scaling up RLVR data by exploiting abundant, reasoning-rich, unverifiable internet text.

## 1. Introduction

Reinforcement Learning with Verifiable Rewards (RLVR) has emerged as a core ingredient for unlocking complex reasoning behavior in Large Language Models (LLMs), driving the recent breakthrough of frontier reasoning models such as DeepSeek-R1 (Guo et al., 2025a), OpenAI-o3 (OpenAI, 2025) and Gemini-3 (Google DeepMind, 2025). Specifically, several recent efforts (Liu et al., 2025a; Hu et al., 2025b;c; Khatri et al., 2025) have focused on scaling up RLVR (e.g., through extended training steps or rollout budgets), aiming to achieve continuous performance gains with increasing compute. While these scaling recipes yield steady initial gains, model improvements increasingly saturate on finite training data (Zeng et al., 2025a; Hu et al., 2025c; Kumar et al., 2024; Khatri et al., 2025).

Scaling up RLVR data is challenging due to the strict format requirements imposed by verifiable reward computation, which limits training data to problems with ground-truth solutions amenable to simple automatic validation, such as math problems parsable by a math verifier, or coding problems with unit tests executable in a sandbox environment. One of the primary approaches in prior work is then to source human-authored verifiable problems (Chen et al., 2025; Luo et al., 2025; Albalak et al., 2025; Cui et al., 2025; Lu et al., 2025; Gao et al., 2024). However, this is expensive, difficult to scale, and limited to narrow domains. As a result, tasks with long-form or open-ended solutions that are hard to automatically verify (e.g., math theorem proving or medical diagnostic reasoning) are typically discarded.

Recent attempts to automatically synthesize RLVR data also rely on human expertise to construct handcrafted verifiable environments (i.e., procedural data generators) that span logical puzzles, math, games and other formal domains (Stojanovski et al., 2025; Lacombe et al., 2025; Zeng et al., 2025a; Xu et al., 2026). Although they enable generating infinite examples with tunable complexity on a fixed environment, it is difficult to scale beyond hundreds of distinct

[1]NVIDIA [2]University of Washington [3]University of California San Diego. Correspondence to: Ximing Lu <lux32@cs.washington.edu>.

*Proceedings of the $43^{rd}$ International Conference on Machine Learning*, Seoul, South Korea. PMLR 306, 2026. Copyright 2026 by the author(s).

## Reasoning-rich Unverifiable Text

### 🛡 Web Scrapes

… log: computed the time to create 4000 empty files in the same directory, but with names chosen to hash to the same CRC32C value. The operation failed after 5 seconds and only 61 files were created. Posted by user · 2023-07-14 (testing on local filesystem, details omitted) An attacker in a shared-directory scenario can prevent a victim from creating a file with a known-in-advance name by exploiting directory hash collisions. Group-writable directories and shared locations like /tmp are especially exposed to this kind of denial-of-service behavior, since any unprivileged user can interfere with ….

### 📗 STEM Textbook

**Ch.6 – Q.17**: When $NH_3$(aq) is added to $CuSO_4$(aq), a dark blue solution forms with a precipitate. Explain the chemical reactions involved in this process.

**A**: When $NH_3$(aq) is added to $CuSO_4$(aq), it acts as a weak base to generate $OH^-$ in solution … The key balanced steps are:
$$NH_3(aq) + H_2O(l) \rightleftharpoons NH_4^+(aq) + OH^-(aq)$$
$$Cu^{2+}(aq) + 2\,OH^-(aq) \rightarrow Cu(OH)_2(s)$$
$$Cu(OH)_2(s) + 4\,NH_3(aq) \rightarrow [Cu(NH_3)_4]^{2+}(aq)$$
$$+2\,OH^-(aq)$$
The precipitate dissolves only when $NH_3$ …

### 🧑‍💻 Coding Problem w/o Test Cases

📄 **user_r2lx**: I tried to make a dynamic 5x5 int array `int **data=malloc(5*5);` But I get segmentation fault on trying to access it.

📄 **user_krylon**: You need to allocate memory for the 2d-array you want to make. First, you have to allocate the space for pointers …

```
int **data=(int**)malloc(sizeof(*data)*5);
for (int r=0; r<5; r++) {
    data[r]=(int*)malloc(sizeof(**data)*5);
}
```

If you want contiguous block of memory for the whole array, you can allocate a single …

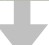 **Large Language Model**

Given a source document, extract or summarize a coherent passage that is educationally valuable. Then, construct **a multiple-choice version of a fill-in-the-middle task** based on this passage. Identify a consecutive multi-sentence span of text that contains several important reasoning steps and replace it with **[MASK]**.

Given a question and its reference solution, construct **a multiple-choice version of a fill-in-the-middle task**. Identify several consecutive steps that are important in the reference solution and replace them with **[MASK]**.

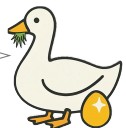

The removed content should serve as the ground-truth answer. Then, generate at least ten diverse **distractors** that are plausible, similar in style and length to the removed content, but incorrect. Return your answer in the following JSON format within <answer></answer>:
{"masked_reference_solution": "...", "removed_steps": "...", "distractors": ["...", "...", "..."]}

📝 **Multiple-Choice Question**

## ⭐ Unlimited RLVR Tasks ⭐

### 🛡 Cybersecurity MCQ

You are given a cybersecurity passage with some content replaced by **[MASK]**. Carefully analyze the context and select the option that best fills in the **[MASK]**.

**Passage:**
I computed the time to create 4000 empty files in the same directory, but with names chosen to hash to the same CRC32C value. The operation failed after 5 seconds and only 61 files were created. **[MASK]** Group-writable directories and shared locations like /tmp are especially exposed to this kind of denial-of-service behavior, since any unprivileged …

**Options:**
(A) Under adversarial conditions, directory operations always degrade to the point of functional denial-of-service …
(B) An attacker in a shared-directory scenario can prevent a victim from creating a file with a known-in-advance name by …
(C) An attacker can prevent a victim from creating any file at all, even in private directories by triggering system-wide …
...
(H) Deliberate directory hash collisions provide a practical mechanism for disrupting file …

### 📗 STEM MCQ

You are given a science problem and its solution, with some steps of the solution replaced by **[MASK]**. Select the option that best fills in the **[MASK]**.

**Question:**
When $NH_3$(aq) is added to $CuSO_4$(aq), a dark blue solution forms with a precipitate. Explain the chemical reactions involved.

**Solution:**
When $NH_3$(aq) is added to $CuSO_4$(aq), it acts as a weak base to generate $OH^-$ in solution … The key balanced steps are:
**[MASK]**
$$Cu^{2+}(aq) + 2\,OH^-(aq) \rightarrow Cu(OH)_2(s)$$
$$Cu(OH)_2(s) + 4\,NH_3(aq) \rightarrow [Cu(NH_3)_4]^{2+}(aq)$$
$$+2\,OH^-(aq)$$
The precipitate dissolves only when $NH_3$ …

**Options:**
(A) $NH_3(aq) + OH^-(aq) \rightleftharpoons NH_2^-(aq) + H_2O(l)$
(B) $NH_3(aq) \rightarrow NH_4^+(aq) + e^-$
(C) $NH_3(aq) + H_2O(l) \rightleftharpoons NH_4^+(aq) + OH^-(aq)$
...
(G) $NH_3(aq) + H_2O(l) \rightleftharpoons NH_4^+(aq) + O^{2-}(aq)$

### 🧑‍💻 Coding MCQ

You are given a coding problem and its solution, with some parts of the solution replaced by **[MASK]**. Select the option that best fills in the **[MASK]**.

**Question:**
I tried to make a dynamic 5x5 int array `int **data=malloc(5*5);` But I get segmentation fault on trying to access it.

**Solution:**
You need to allocate memory for the 2d-array you want to make. First, you have to …

```
int **data=(int**)malloc(sizeof(*data)*5);
for (int r=0; r<5; r++) {
    [MASK]
}
```

If you want contiguous block of memory for the whole array, you can allocate a single …

**Options:**
(A) `data[r]=(int*)malloc(sizeof(int));`
(B) `data[r]=(int*)malloc(sizeof(**data)*5);`
(C) `data[r]=malloc(5);`
...
(H) `data[r]=(int*)malloc(sizeof(*data)*5);`

*Figure 1.* **The 𝒢olden 𝒢oose pipeline.** We synthesize RLVR tasks from unverifiable text by constructing a MCQ version of the fill-in-the-middle task. Given a source text, we prompt an LLM to first identify a contiguous span of crucial reasoning steps and replace it with a `[MASK]`, treating the removed content as the ground-truth answer, and then generate a set of diverse distractors that are plausible and similar to the masked span, yet incorrect. For noisy data sources (e.g., web scrapes), we prompt the LLM to first extract an educationally valuable passage and then construct the MCQ based on it. We further apply difficulty-based filtering to remove easy problems.

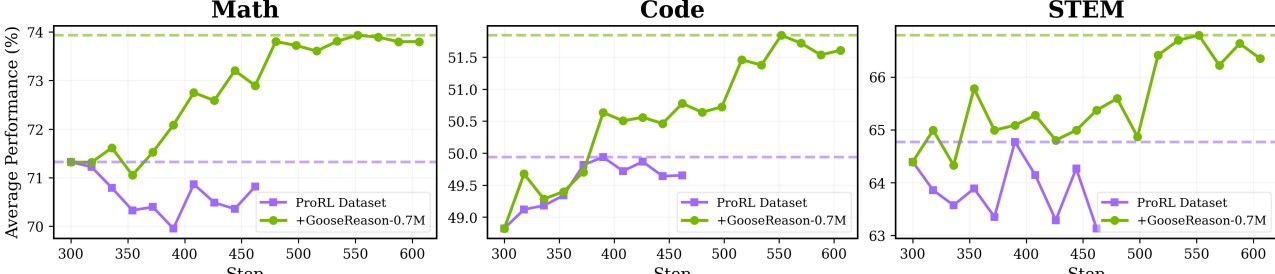

*Figure 2.* Comparison of continued RL training on Qwen-4B-Instruct after data saturation using the original **ProRL data** versus **adding GooseReason-0.7M**. The former exhibits performance plateaus or regression, while the latter yields robust, continuous gains.

environments due to the reliance on manual design. Furthermore, the high-level reasoning patterns in the logical problems generated from these handcrafted environments often resemble those found in human-sourced verifiable problems, consistently excluding open-ended reasoning tasks that are hard to automatically verify.

To tackle these challenges, we introduce 𝒢olden 𝒢oose 🦢, a simple trick to synthesize *unlimited RLVR tasks* from unverifiable internet text by constructing a multiple-choice question-answering version (MCQ) of the fill-in-the-middle task, as shown in Figure 1. Concretely, given a source text[1], we prompt an LLM to first identify a contiguous span of crucial reasoning steps and replace it with a `[MASK]`, treating the removed content as the ground-truth answer, and then generate a set of diverse distractors that are plausible and similar in style to the masked span, yet incorrect. Notably, this enables us to leverage reasoning-rich unverifiable corpora that were typically excluded from prior RLVR data construction, including Olympiad-level theorem proving from AoPS-Instruct (Mahdavi et al., 2025b), free-form textbook QA from MegaScience (Fan et al., 2025), and coding problems lacking test cases from rStar-Coder (Liu et al., 2025b). From these sources, we construct 𝒢ooseReason-0.7M, a large-scale RLVR dataset comprising over 0.7 million tasks spanning mathematics, programming, and general scientific domains, to effectively complement existing RLVR datasets and enable RL to scale further, while remain seamlessly pluggable into any RL recipe.

Empirically, we show that *GooseReason-0.7M* effectively scales up RL training beyond the data saturation point of existing RLVR datasets. For one of the current strongest 1.5B RL-trained LMs, `ProRL-1.5B-v2` (Hu et al., 2025b), which was originally trained large-scale using the ProRL recipe (Liu et al., 2025a) for over 20,000 H100 GPU hours performance saturates upon further training with the same recipe (Hu et al., 2025c; Zeng et al., 2025b). As shown in Figure 3, only around 25% of the 136k RLVR samples used in ProRL training remain effective at this point, with the rest becoming stale, where the model consistently succeeds or fails across all rollouts, providing no learning signal. By in-

corporating fresh RLVR samples from *GooseReason-0.7M*, we observe robust, continuous performance gains over an additional 1,100 H100 GPU hours of training across 15 diverse benchmarks covering mathematics, code generation, STEM, and logical reasoning, whereas continuing with the original ProRL data yields negligible improvement (Figure 5). We see the biggest difference in the STEM domain (an absolute gain of 3.48% versus 0.13%), as existing RLVR data in the general science domain is much scarcer than for math and code—a gap that *GooseReason* substantially bridges.

More importantly, we find that data saturation occurs much earlier and is more severe with stronger LLMs. While the ProRL recipe manages to train DeepSeek-R1-1.5B for over two thousand steps with continuous gains, when we apply the same recipe to `Qwen-4B-Instruct` (Team, 2025), performance plateaus or even degrades after merely 300 steps. *GooseReason* effectively revives the saturated model (Figure 2), enabling continuous RL training with an absolute improvement of 2.27% (versus prior 0.79% degradation). The resulting model, 𝒢ooseReason-4B-Instruct, achieves **new state-of-the-art performance among 4B-Instruct models across 15 diverse benchmarks**. Interestingly, *GooseReason* also drives performance gains on downstream tasks whose domains are not explicitly covered by its data, such as logical puzzles, indicating improved reasoning generalization. Furthermore, we find that *GooseReason* enables more efficient RL scaling under a fixed compute budget (Figure 6). We train Qwen-4B-Instruct from scratch for only 200 steps with ProRL data alone versus joint training with *GooseReason-0.7M*, and find the latter consistently achieves higher performance at the same number of steps.

Finally, we deploy 𝒢olden 𝒢oose in a real-world setting and synthesize RLVR data for cybersecurity, a specialized domain where open-source RLVR data is non-existent. By leveraging cybersecurity-related web scrapes primarily from FineWeb (Yu et al., 2025), we constructed **𝒢ooseReason-Cyber** with 180K RLVR examples. Training Qwen-4B-Instruct on this data for a mere 100 RL steps yields a 4.44% absolute gain across 3 cybersecurity benchmarks, establishing a new state-of-the-art for cybersecurity LLMs. In contrast, the previous SOTA, Llama-Primus-Instruct (Yu et al., 2025), achieved an average gain of only 1.44% over its base

---

[1]Our source corpora consist of QA pairs for the reasoning domain and raw web scrapes for the cybersecurity domain.

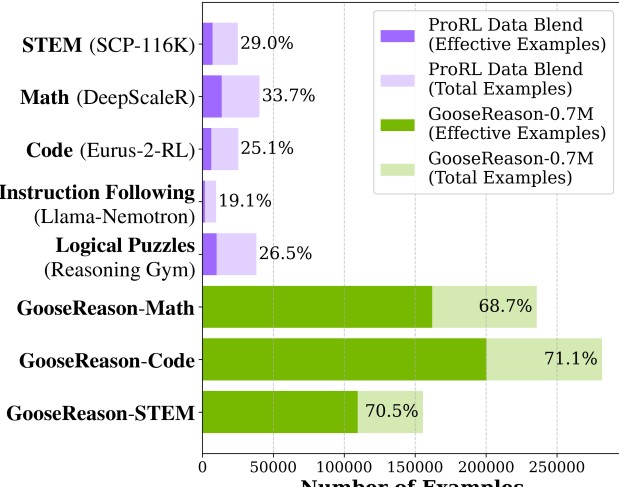

*Figure 3.* Comparison between *GooseReason-0.7M* and existing RLVR datasets used in ProRL (Liu et al., 2025a) in terms of total examples and effective examples, measured relative to ProRL-1.5B-v2. We define an example as effective if it has both successful and failed model rollouts, yielding meaningful learning signal for RL. Notably, we increase the number of effective examples in math, code, and STEM by over 450,000, which is a **13×** increase over the total effective examples in the ProRL dataset.

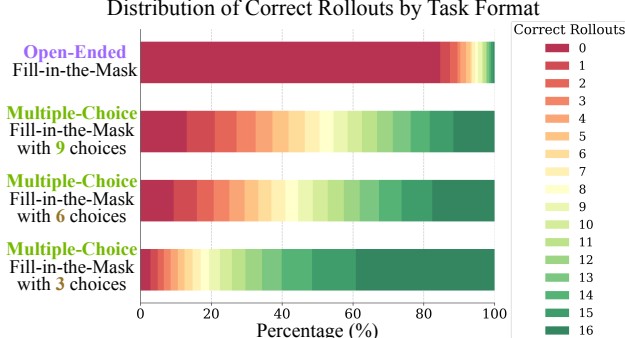

*Figure 4.* Accuracy distribution of `ProRL-1.5B-v2`, calculated as the success rate over 16 rollouts per task, on *GooseReason-Math* across different task formulations. Notably, with 9-choice MCQ format, the majority of problems fall into a medium-difficulty regime (exhibiting both successful and failed model rollouts), providing the most effective signals for RL training.

model (Llama-3.1-8B-Instruct), despite undergoing extensive domain-specific pre-training and post-training. These results highlight $\mathcal{G}$olden $\mathcal{G}$oose as a scalable path for transforming abundant, reasoning-rich, yet unverifiable internet text into high-quality RLVR tasks that fuel RL scaling.

## 2. Method: $\mathcal{G}$olden $\mathcal{G}$oose

### 2.1. Data Synthesis Pipeline

As illustrated in Figure 1, given a source text $S$, we prompt an LLM to identify a contiguous span $t$ of important reasoning steps, which is used to construct a masked context $S_{\text{mask}}$ by replacing $t$ in $S$ with a special token [MASK]. Treating $t$ as the ground-truth answer, the LLM then generates a set of diverse distractors $\mathcal{D} = \{d_1, d_2, \ldots, d_k\}$ that are plausible

and similar in style to $t$, yet incorrect in the context of $S_{\text{mask}}$. Finally, we formulate a multiple-choice question $\mathcal{Q}$

$$\mathcal{Q} = (S_{\text{mask}}, \{t\} \cup \mathcal{D})$$

If the source text $S$ is noisy, such as cybersecurity-related scrapes from FineWeb, we prompt the LLM to first extract or summarize $S$ into a coherent, educationally valuable $S'$, and then construct $S_{\text{mask}}$ and $\mathcal{D}$ based on $S'$. If $S$ contains no suitable passage, the LLM is instructed to return an empty string. The student model is provided with $S_{\text{mask}}$ and tasked with selecting the option that best fills the [MASK] from the candidate set $\{t\} \cup \mathcal{D}$, presented in randomized order. Verification during RL simply checks if the prediction matches the ground-truth option. See Appendix A for the prompts used in data synthesis and question formulation.

To ensure data quality, we used the strongest LLM available at the time of the experiment, GPT-5 (OpenAI, 2025), for the synthesis pipeline. For reasoning-dense source text (e.g., AoPS-Instruct, rStar-Coder, MegaScience), we found the questions constructed by GPT-5 were of sufficient quality and difficulty to require no further post-processing. For noisy source text (e.g., FineWeb), we found some masked spans could be easily inferred from context rather than requiring reasoning; thus, we additionally employ difficulty-based filtering to remove easy problems on which the student model consistently succeeds across all 16 rollouts.

### 2.2. Source Corpora

#### 2.2.1. REASONING DOMAIN

We leverage existing reasoning-rich, unverifiable corpora that were typically excluded from previous RLVR data curation to construct *GooseReason-0.7M.*

**AoPS-Instruct**  Mahdavi et al. (2025a) extracted around 600k question-answer pairs from the Art of Problem Solving (AoPS) forum, which predominantly features Olympiad-level math problems and community-driven solutions. Due to the unstructured and noisy nature of the forum, solutions often vary in format and style, and are occasionally incomplete. Additionally, AoPS contains a large number of theorem-proving problems whose solutions consist of entire math proofs, which are impossible to verify with a math verifier under existing RLVR pipeline.

**rStar-Coder**  Liu et al. (2025b) curated and cleaned 37.7K expert-written problems with oracle solutions from competitive programming platforms (e.g., IOI, Codeforces) and use them as seeds to synthesize new problems. They also proposed an input-output test case synthesis pipeline consisting of a three-step input generation method and a mutual verification mechanism for output labeling. However, only 380K out of 1,656K synthesized questions successfully obtained test cases through this pipeline. In the released data, the

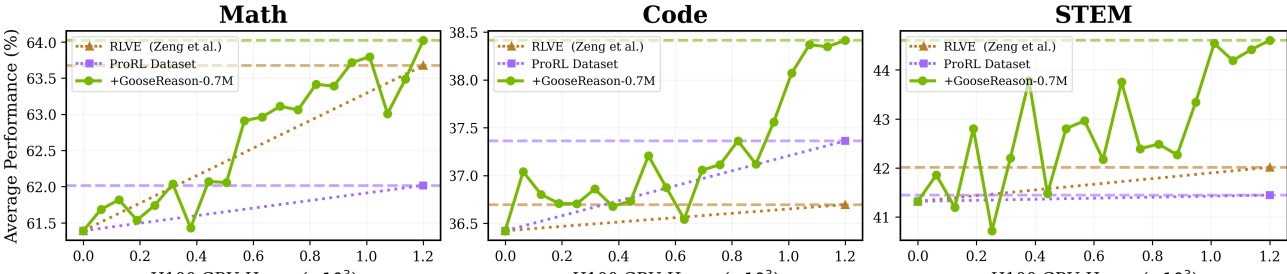

*Figure 5.* Comparison of continued RL training on ProRL-1.5B-v2 using the original **ProRL data**, adding **GooseReason-0.7M**, or using **RLVE** (Zeng et al., 2025a). Continuing with ProRL data yields marginal gains, adding GooseReason-0.7M produces robust, continuous improvements, while RLVE is highly effective in math but less so in STEM and coding.

`synthetic_sft` split contains only questions and teacher model's solutions without test cases, and is therefore not directly usable for RL training; we leverage this split to synthesize verifiable coding questions with *Golden Goose.*

**MegaScience**  Fan et al. (2025) exacted 650k question-answer pairs from nearly 12k university-level scientific textbooks spanning various subjects, including physics, biology, chemistry, medicine, computer science, mathematics, and economics. Many solutions in domains such as chemistry involve specialized scientific formulas, while many questions in domains like medicine or economics are free-form or open-ended, requiring multi-paragraph discussions or explanations. Both are challenging to validate under the verifier-based approach in current RLVR pipeline.

From these sources, we synthesized over 0.7 million novel RLVR tasks with *Golden Goose* pipeline. Figure 3 compares *GooseReason* with existing RLVR datasets used in ProRL (Liu et al., 2025a) in terms of total examples and effective examples relative to a heavily RL-trained model, `ProRL-1.5B-v2`. We find that only about 25% of the 136K samples in the ProRL data blend provide meaningful learning signals for continual RL, eliciting both successful and failed model rollouts. In contrast, *GooseReason-0.7M* retains around 70% effectiveness ratio, substantially supplementing existing RLVR datasets to further scale RL training.

### 2.2.2. CYBERSECURITY DOMAIN

Unlike the reasoning domain, highly specialized domains such as cybersecurity lack open-source RLVR datasets. Primus (Yu et al., 2025) released the pre-training data for their cybersecurity LLM, `Llama-Primus-Instruct`, which consists of two components: **Primus-Seed**, comprising data crawled from reputable sources such as MITRE, Wikipedia, and well-known cybersecurity company websites, as well as cyber threat intelligence (CTI) manually collected by threat experts; and **Primus-FineWeb**, constructed by filtering cybersecurity-related text from FineWeb using Primus-Seed as positive samples. These data sources are primarily web scrapes and are therefore extremely noisy. We deployed *Golden Goose* in the wild and synthesized approximately 180K RLVR tasks for the cybersecurity domain

out of raw internet text.

### 2.3. Design Choice

**Multiple-Choice v.s. Open-ended**  An alternative to the multiple-choice formulation is to construct RLVR tasks as open-ended fill-in-the-mask problems, where the model is tasked with predicting the masked content and an LLM-as-judge verifies the prediction against the ground-truth. However, beyond the computational overhead of hosting a powerful judge model during RL training, we observe that reasoning models, particularly those heavily tuned with RL, exhibit a strong tendency to solve the problem from scratch and completely ignore the task requirement of generating the infill. As shown in Figure 4, over 83% of examples in the open-ended version of *GooseReason-Math* result in consistent zero accuracy for `ProRL-1.5B-v2`, yielding no usable RL signal, largely due to poor instruction following.

**Number of Distractors**  We ablate the effect of the number of distractors, as shown in Figure 4. With too few options (e.g., 3), the majority of problems in *GooseReason-Math* become overly easy for `ProRL-1.5B-v2`, where the model tends to rely on an elimination strategy—identifying flaws in the provided options—rather than performing the intended reasoning to infer the masked content. Increasing the number of distractors raises the task difficulty, as this elimination strategy becomes less effective under a fixed output length. When using 9 options, over 70% of the problems fall into a medium-difficulty regime with both successful and failed model rollouts, effective for RL training.

## 3. Experiment

### 3.1. Scaling Up RL Training via *GooseReason-0.7M*

We evaluate the effect of *GooseReason-0.7M* across two representative scenarios for scaling up RL training of LLMs. First, we consider a data-saturation scenario, where the model has already saturated on a strong RLVR data blend (§ 3.1.2). Second, we study a compute-constrained scenario, where RL training starts from scratch under a fixed training budget, making the choice of RL data crucial (§ 3.1.2).

*Table 1.* Performance (pass@1) comparison across math benchmarks. While RL training using ProRL data yields substantial initial gains, performance plateaus or degrades after 300 steps; adding GooseReason-0.7M revives the saturated model and enables further RL scaling. The results of Qwen3-30B-Instruct are marked as gray and are provided as a reference

| Model | RL Data | RL Steps | AIME24 | AIME25 | AMC | MATH | Minerva | Olympiad | Avg |
|---|---|---|---|---|---|---|---|---|---|
| Qwen3-4B-Instruct | N/A | | 64.79 | 48.75 | 85.17 | 94.66 | 50.09 | 65.83 | 68.21 |
| | ProRL Dataset | 333 | 66.46 | 57.29 | 87.80 | 96.41 | 53.72 | 68.24 | 71.65 |
| | | +156 | 62.29 | 55.21 | 87.65 | 96.54 | 53.33 | 67.19 | 70.36 |
| | + GooseReason-0.7M | +270 | **70.00** | **63.96** | **89.16** | **96.70** | **54.37** | **68.79** | **73.83** |
| Qwen3-30B-Instruct | N/A | | 76.66 | 63.74 | 91.64 | 97.10 | 51.99 | 70.05 | 75.20 |

*Table 2.* Performance (pass@1) comparison across coding benchmarks.

| Model | RL Data | RL Steps | APPS | CodeContests | CodeForces | TACO | HumanEvalPlus | LiveCodeBench | Avg |
|---|---|---|---|---|---|---|---|---|---|
| Qwen3-4B-Instruct | N/A | | 47.01 | 42.08 | 33.69 | 23.69 | 77.56 | 31.74 | 42.63 |
| | ProRL Dataset | 333 | 57.92 | 52.55 | 51.67 | 33.13 | 84.24 | 41.28 | 53.46 |
| | | +156 | 58.45 | 52.88 | 54.47 | 32.80 | 84.20 | 40.56 | 53.89 |
| | +GooseReason-0.7M | +270 | **60.48** | **54.66** | **55.59** | **35.37** | **86.46** | **41.64** | **55.70** |
| Qwen3-30B-Instruct | N/A | | 55.37 | 49.70 | 47.76 | 29.05 | 80.56 | 43.20 | 50.94 |

*Table 3.* Performance (pass@1) comparison on STEM reasoning (GPQA Diamond), instruction following (IFEval), and logic puzzles (Reasoning Gym). Tasks in Reasoning Gym are grouped into four primary categories: Math (algebra, arithmetic, geometry, graphs), Algorithmic (algorithmic, code), Cognition (arc, games, cognition) and Logic (logic, induction).

| Model | RL Data | RL Steps | GPQA | IFEval | Math | Algorithmic | Cognition | Logic | Avg. Gym |
|---|---|---|---|---|---|---|---|---|---|
| Qwen3-4B-Instruct | N/A | | 60.26 | 72.36 | 43.69 | 19.46 | 34.92 | 57.26 | 33.98 |
| | ProRL Dataset | 333 | 64.39 | 76.11 | 92.66 | 80.47 | 60.07 | 86.90 | 80.10 |
| | | +156 | 62.87 | 76.24 | 92.71 | 83.24 | **60.75** | 87.71 | 81.06 |
| | +GooseReason-0.7M | +270 | **66.79** | **76.39** | **92.76** | **83.91** | 60.24 | **87.80** | **81.28** |
| Qwen3-30B-Instruct | N/A | | 70.40 | 82.73 | 53.86 | 38.51 | 28.60 | 32.89 | 43.56 |

**RL Algorithm** *GooseReason* is compatible with any RL algorithm applicable to RLVR. In this work, we adopt the RL recipe in ProRLv2 (Hu et al., 2025b), which is a variant of the GRPO algorithm (Shao et al., 2024) designed to maintain stable policy optimization over prolonged training. Specifically, it employs the clipped GRPO objective with a decoupled advantage normalization strategy from REINFORCE++ (Hu et al., 2025a) consisting of a group-wise mean subtraction followed by batch-level standardization.

**Evaluation** Following ProRL, we evaluate models on 15 benchmarks in various domains. Math performance is tested on AIME 2024/2025 (MAA, 2024; 2025), AMC (MAA), MATH (Hendrycks et al., 2021b), Minerva (Lewkowycz et al., 2022), and Olympiad Bench (He et al., 2024). Coding is assessed using the PRIME validation set (Cui et al., 2025), covering APPS (Hendrycks et al., 2021a), CodeContests (Li et al., 2022), CodeForces, and TACO (Li et al., 2023), alongside HumanEvalPlus (Liu et al., 2023) and LiveCodeBench (Jain et al., 2024). STEM reasoning is measured through GPQA Diamond (Rein et al., 2023), logical reasoning via Reasoning Gym (Stojanovski et al., 2025), and instruction following via IFEval (Bae et al., 2025).

### 3.1.1. SCALING BEYOND DATA SATURATION

We first evaluate whether *GooseReason-0.7M* can drive further scaling in a saturated model that has undergone prolonged RL training. Specifically, we start from one of the strongest open-source RLVR-ed models, ProRL-1.5B-v2 (Hu et al., 2025b), which was originally trained from R1-Distill-Qwen-1.5B (Guo et al., 2025b) using over 20K H100 GPU hours, and has reached performance saturation (Hu et al., 2025c) on a 136K diverse training data blend spanning mathematics, coding, logical reasoning, STEM, and instruction-following.

As shown in Figure 5, continued RL with the original ProRL data blend yields marginal improvements over 1,100 H100 GPU hours. In contrast, incorporating *GooseReason-0.7M* revives the saturated model and leads to robust, continuous performance gains across all domains: 2.71% versus 0.63% in math, 2.12% versus 0.95% in coding, and a notable 3.48% versus 0.13% in STEM. The margin is largest in STEM, where *GooseReason* bridges the scarcity of general science RLVR data relative to the more abundant math and code domains. Importantly, despite the MCQ format of *GooseReason*, the evaluation targets primarily non-MCQ benchmarks, suggesting that the model acquires generalizable reasoning skills that transcend a specific task format. We further compare against RLVE (Zeng et al., 2025a), using their publicly released checkpoint trained under an equivalent computational budget. While RLVE is highly effective on math, its impact on STEM is limited to a 0.62% gain. While synthetic RL environments excel at algorithmic tasks like math and code, it remains unclear how to adapt such procedural generation to knowledge-intensive STEM domains like medicine, economics and cybersecurity.

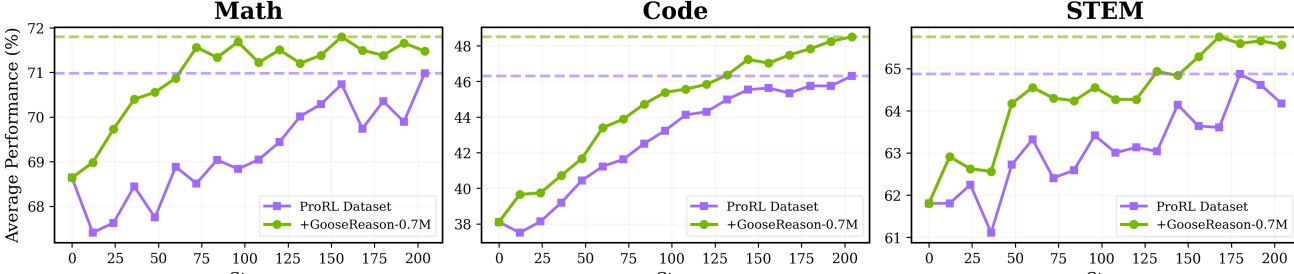

*Figure 6.* Comparison of RL training from scratch on Qwen-4B-Instruct under a fixed compute budget with **ProRL data only** versus **joint training with GooseReason-0.7M**. The latter consistently achieves higher performance at the same number of steps.

Furthermore, we find that data saturation occurs much earlier and is more severe with stronger LLMs. While the ProRL recipe enables continuous gains for R1-1.5B over 2K steps, applying the same recipe to `Qwen-4B-Instruct` (Team, 2025) results see a performance plateau or even degradation after merely 300 steps. As shown in Figure 2 and Table 1, further training leads to a 1.29% loss in math, a marginal 0.43% gain in coding, and a 1.52% loss in STEM. In contrast, incorporating *GooseReason* reverses this trend with robust absolute gains of 2.18%, 2.24%, and 2.40%, respectively. Interestingly, *GooseReason* also enables further improvement on downstream tasks not directly covered by its data, such as logical puzzles in Reasoning Gym, indicating the transferability of the acquired reasoning skills. The resulting model, *GooseReason-4B-Instruct* achieves new state-of-the-art results among 4B-Instruct models across 15 diverse benchmarks. Even compared to a 7.5× larger model, Qwen3-30B-Instruct, our model achieves comparable or even better performance across the board.

We also compare scaling behavior of Qwen-4B-Instruct across various tasks in continued RL with and without *GooseReason-0.7M* (Figure 7), and group them into three categories: *diverge* (regression vs. gain), *outpace* (faster gains), and *align* (similar trends). We find that STEM and most math tasks fall into the *diverge* category, while coding tasks primarily *outpace*, with a few *diverge* or *align*.

### 3.1.2. COMPUTE-EFFICIENT SCALING

Next, we evaluate whether *GooseReason-0.7M* enables more effective RL scaling under a fixed compute budget. Specifically, we train `Qwen-4B-Instruct` from scratch for only 200 RL steps, comparing training with the ProRL data alone to joint training with *GooseReason-0.7M*. As shown in Figure 6, incorporating *GooseReason-0.7M* consistently achieves higher performance at the same number of steps, enabling more compute-efficient scaling.

### 3.2. RLVR for Cybersecurity via *GooseReason-Cyber*

Finally, we evaluate whether *GooseReason-Cyber* enables RLVR to improve model reasoning capabilities a specialized domain, cybersecurity. Following Yu et al. (2025), we evaluate on 3 cybersecurity benchmarks: CTI-Bench

*Table 4.* Performance comparison on cybersecurity benchmarks between 8B domain-specialized Primus models and a 4B general reasoning model, Qwen3-Instruct, trained with *GooseReason-Cyber*.

| Model | CTI-MCQ | CyberMetric | SecEval | Avg |
|---|---|---|---|---|
| Llama-3.1-8B-Instruct | 64.20 | 85.60 | 49.66 | 66.49 |
| Llama-Primus-Instruct | 66.60 | 86.40 | 49.43 | 67.48 |
| Llama-Primus-Merged | 66.56 | 86.60 | 50.62 | 67.93 |
| Qwen3-4B-Instruct | 63.44 | 89.78 | 70.44 | 74.55 |
| GooseReason-Cyber-4B | **73.79** | **92.05** | **71.14** | **78.99** |

(Alam et al., 2024), which assesses threat-intelligence reasoning and vulnerability analysis; CyberMetricc (Tihanyi et al., 2024), which tests knowledge in domains like compliance and penetration testing; and SecEval (Busch et al., 2014), which evaluates proficiency across foundational areas such as software and network security. As shown in Table 4, training `Qwen-4B-Instruct` on *GooseReason-Cyber* for a mere 100 RL steps yields a 4.44% absolute gain across 3 benchmarks, establishing a new state-of-the-art for cybersecurity LLMs. In contrast, the previous SOTA, Llama-Primus-Instruct, achieved an average gain of only 1.44% over its base model (Llama-3.1-8B-Instruct), despite undergoing extensive domain-specific pre-training and post-training. These results underscores the effectiveness of RLVR in specialized domains when fueled by scalable data.

## 4. Related works

**Scaling RLVR.** A central challenge in RLVR is identifying effective axes along which training can be scaled successfully to avoid saturation (Tan et al., 2025; Khatri et al., 2025). Algorithmically, ProRL (Liu et al., 2025a) proposes using a mixture of data containing several reasoning tasks alongside modifications to GRPO to allow training for a longer number of steps. Meanwhile, BroRL proposes to continue scaling by increasing the number of rollouts per example (Hu et al., 2025c). More recently, (Khatri et al., 2025) conducted a large-scale analysis comparing different recipes and proposed ScaleRL leveraging the insights of the analysis. In this work, we take a data-centric perspective. We leverage existing algorithmic insights and propose a simple method to synthesize RLVR data from unverifiable reasoning-rich internet text, effectively complementing existing RLVR datasets and allowing training beyond existing

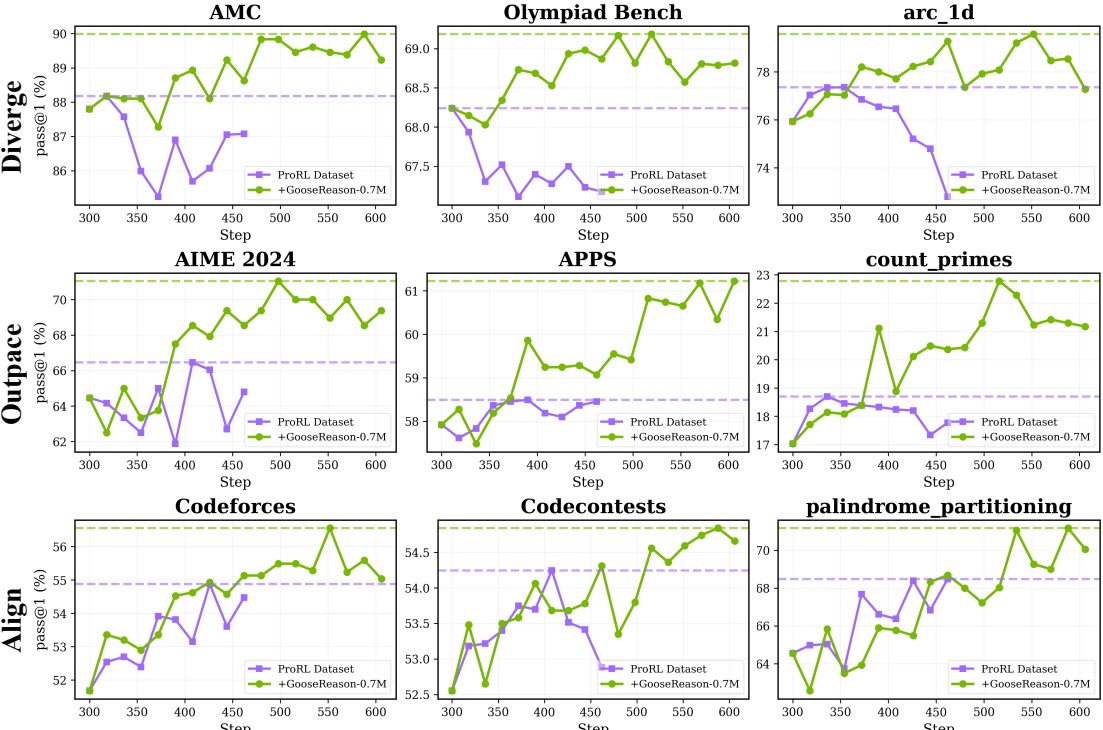

*Figure 7.* Scaling behavior of continued RL training on Qwen-4B-Instruct with **ProRL data only** versus **joint training GooseReason-0.7M**, categorized as *diverge* (regression vs. gain), *outpace* (faster gains), and *align* (similar trends).

algorithms' saturation points.

**Large Scale Human Annotation for RLVR.** Significant effort has been invested by the community to collect large-scale RLVR datasets where curation and verification are conducted by specialized human experts. For instance, Albalak et al. (2025); Gao et al. (2024); Chen et al. (2025); Luo et al. (2025); Cui et al. (2025); Lu et al. (2025) and Jain et al. (2024); Liu et al. (2025b) provide curated and verified large-scale RLVR data for math and code domain, respectively. Our work complements those datasets by focusing on transforming reasoning-rich unverifiable internet text into RLVR tasks without the need for domain experts or handcrafted environments.

**Automated Data Synthesis for RLVR.** Recent attempts to automatically synthesize RLVR data rely on expert-handcrafted verifiable environments. For instance, (Lacombe et al., 2025; Stojanovski et al., 2025) procedurally generate RLVR data using hardcoded environments that span games, puzzles, and formal domains. While (Zeng et al., 2025a) enabled the generation of RLVR data with adaptive problem difficulty for a specific target policy, also leveraging procedural generation within manually engineered environments. More recently (Xu et al., 2026) proposed automatically generating reasoning environments with controllable complexity by transforming programming problems. Our work complements this direction; however, rather than handcrafting environments for procedural gen-

eration or relying on programming problems, we design a simple and scalable pipeline that converts reasoning-rich unverifiable internet text into RLVR data. Notably, this enables the use of unverifiable corpora typically excluded from prior RLVR datasets and gyms, such as free-form textbooks, and coding problems lacking unit tests.

## 5. Conclusion

In this paper, we introduce 𝒢olden 𝒢oose, a simple yet scalable pipeline that unlocks the vast potential of reasoning-rich unverifiable internet text for RLVR by converting it into verifiable multiple-choice tasks. We also release 𝒢ooseReason-0.7M, a large-scale RLVR dataset with over 0.7 million tasks spanning mathematics, programming, and general scientific domains. Our approach effectively revives saturated models, driving sustained performance gains across math, coding, and STEM where standard training recipes previously stagnated, and achieving new SoTA results for 1.5B and 4B-Instruct models across 15 benchmarks. Furthermore, we validate our method's versatility by synthesizing RLVR tasks from raw web scrapes for a specialized domain cybersecurity and establish new SoTA performance that surpasses a 7B domain-specialized model. Our work highlights the potential of automatically re-utilizing reasoning-rich unverifiable internet text to enable RL scaling. Looking forward, we envision this paradigm extending to other high-value disciplines such as law and medicine, where verifiable data is scarce but professional literature is abundant.

## Impact Statement

Our work has the potential to significantly accelerate the progress in reasoning LLMs, particularly in reasoning intensive domains where verifiable RLVR data is scarce such as STEM, math theorem proving, and open-ended domains. A key application demonstrated in this paper is the use of 𝒢olden 𝒢oose in the cybersecurity domain, where we establish new state-of-the-art results. We acknowledge the dual-use nature of this domain; while our goal is to show the versatility of our method and ultimately bolster automated defense and vulnerability analysis, such capabilities could theoretically be misused for offensive operations. Additionally, because our pipeline relies on reasoning-rich internet text, potential biases or toxic content present in the source corpora may be inherited.

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

## A. Details of Data Synthesis

---

**Data Synthesis Prompt for Cybersecurity Domain**

Given a source document, extract or summarize a coherent passage (around 100-600 words) that is educationally valuable for learning Cybersecurity. Then, construct a multiple-choice version of a fill-in-the-middle task based on this passage.  Identify a consecutive multi-sentence span of text that contains several important reasoning steps and replace it with [MASK]. The removed content should serve as the ground-truth answer.  Then, generate at least ten diverse distractors that are plausible, similar in style and length to the removed content, but incorrect.  Return your answer in the following JSON format within <answer></answer>:

```
{
  "masked_passage": "...",
  "removed_content": "...",
  "distractors": [...]
}
```

If there is no suitable passage that is educationally valuable for learning Cybersecurity, return an empty string within <answer></answer>.

**Document:**
[Document]

---

**Data Synthesis Prompt for Math and STEM Domain**

Given a question and its reference solution, construct a multiple-choice version of a fill-in-the-middle task.  Identify several consecutive steps that are important in the reference solution and replace them with [MASK]. The removed steps should serve as the ground-truth answer.  Then, generate at least ten diverse distractors that are plausible, similar in style and length to the removed steps, but incorrect.  Return your answer in the following JSON format within <answer></answer>:

```
{
  "masked_reference_solution": "...",
  "removed_steps": "...",
  "distractors": [...]
}
```

**Question:**
[Question]

**Reference Solution:**
[Solution]

---

**Data Synthesis Prompt for Code Domain**

```
Given a coding question and its reference solution, construct a
multiple-choice version of a fill-in-the-middle task.  Identify several
consecutive lines of code that are important in the reference solution and
replace them with [MASK]. The removed lines should serve as the ground-truth
answer.  Then, generate at least ten diverse distractors that are plausible,
similar in style and length to the removed lines, but incorrect.  Return your
answer in the following JSON format within <answer></answer>:
```

```
{
  "masked_reference_solution": "...",
  "removed_lines": "...",
  "distractors": [...]
}
```

**Question:**
```
[Question]
```

**Reference Solution:**
```
[Solution]
```

# B. Details of Experiments

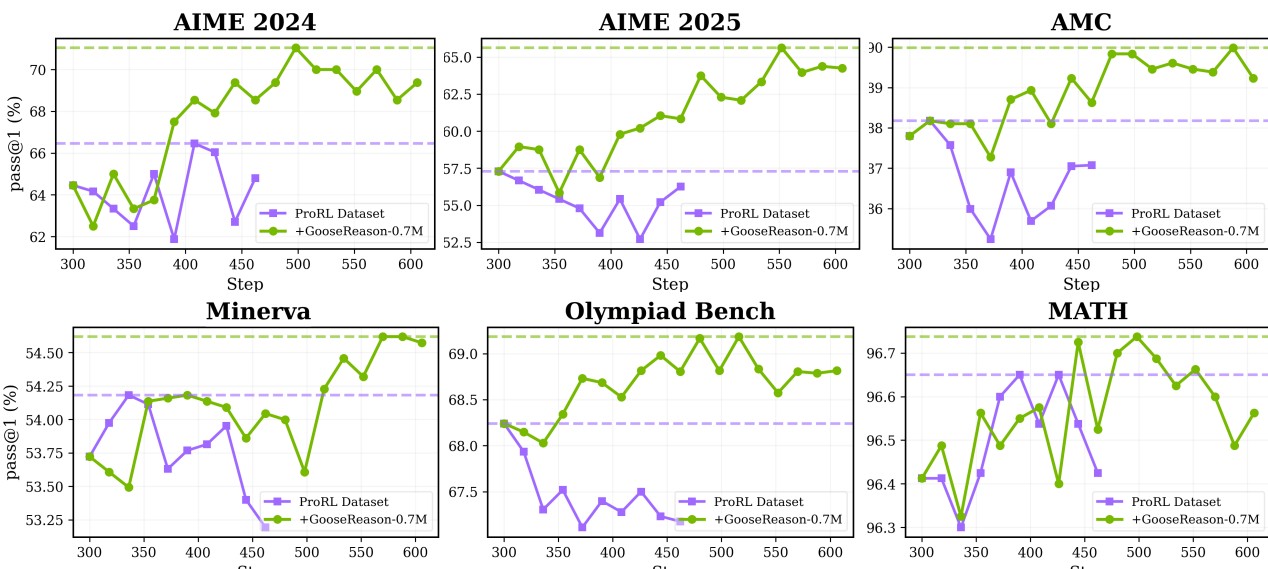

*Figure 8.* Results breakdown for Figure 2 on six math benchmarks: comparison of continued RL training on Qwen-4B-Instruct after data saturation using the original **ProRL data** versus **adding GooseReason-0.7M**.

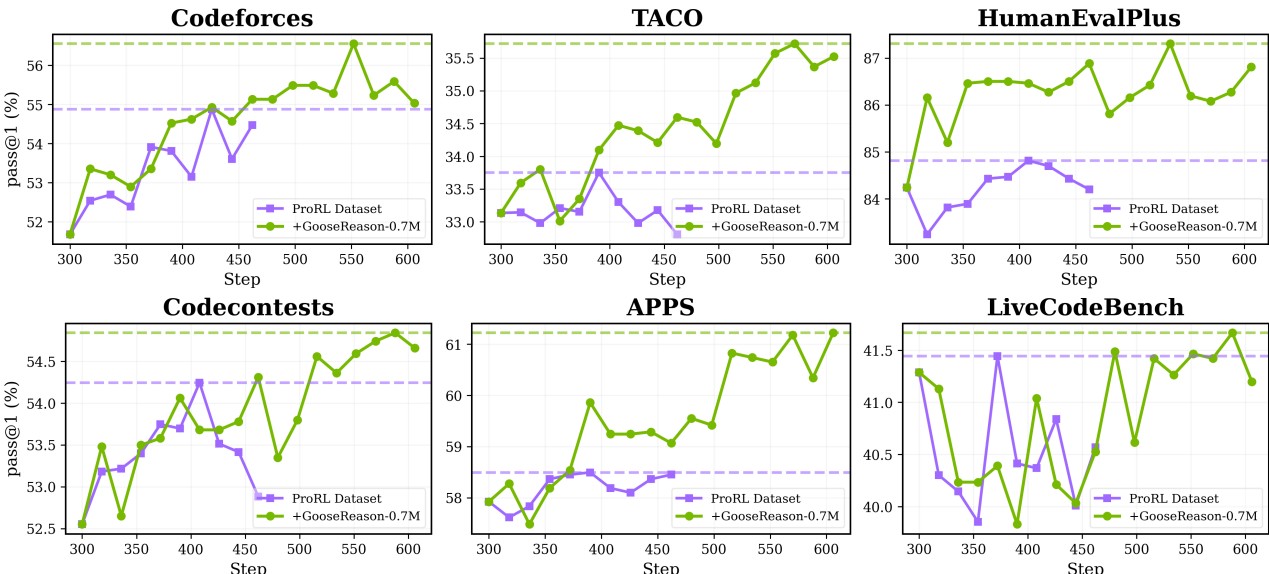

*Figure 9.* Results breakdown for Figure 2 on six coding benchmarks: comparison of continued RL training on Qwen-4B-Instruct after data saturation using the original **ProRL data** versus **adding GooseReason-0.7M**.

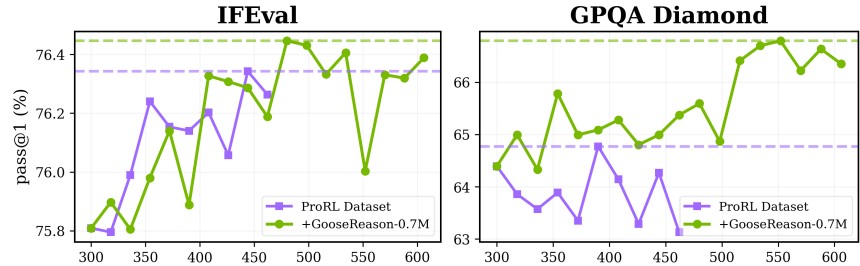

*Figure 10.* Additional results for Figure 2 on IFEval and GPQA Diamond: comparison of continued RL training on Qwen-4B-Instruct after data saturation using the original **ProRL data** versus **adding GooseReason-0.7M**.

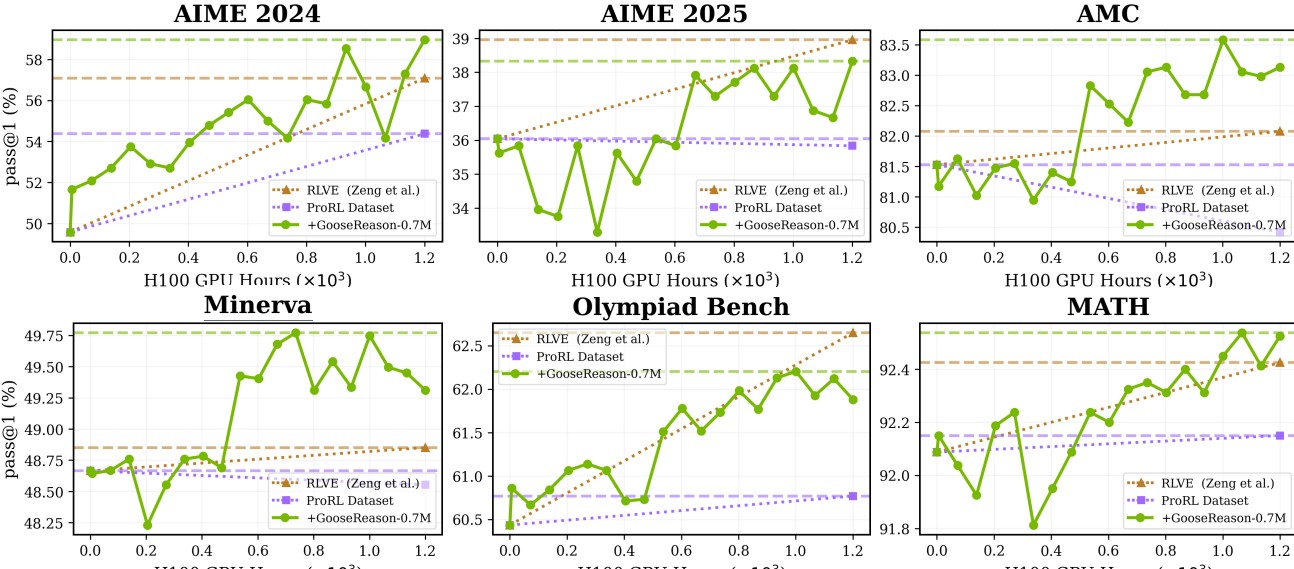

*Figure 11.* Results breakdown for Figure 5 on six math benchmarks: comparison of continued RL training on ProRL-1.5B-v2 using the original **ProRL data**, **adding GooseReason-0.7M**, or using **RLVE**. (Zeng et al., 2025a)

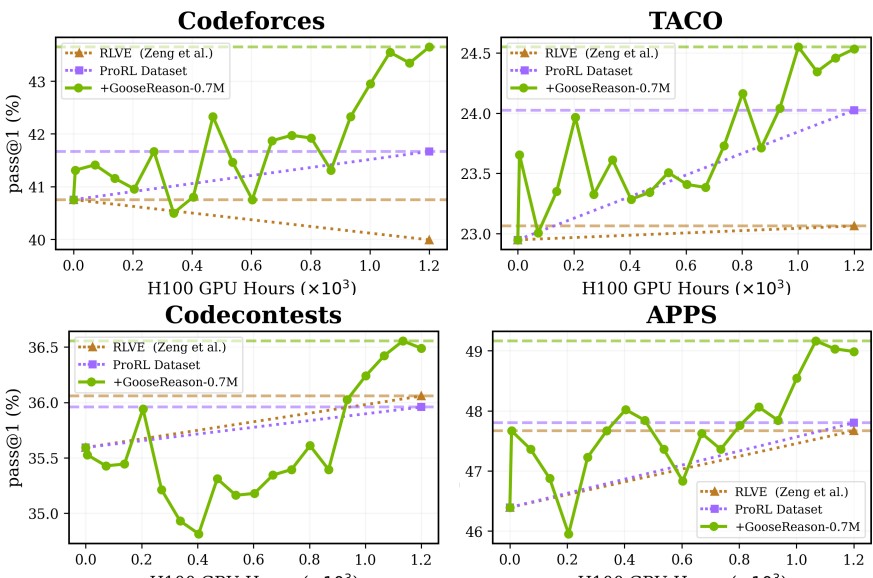

*Figure 12.* Results breakdown for Figure 5 on four coding benchmarks: comparison of continued RL training on ProRL-1.5B-v2 using the original **ProRL data**, **adding GooseReason-0.7M**, or using **RLVE**. (Zeng et al., 2025a)

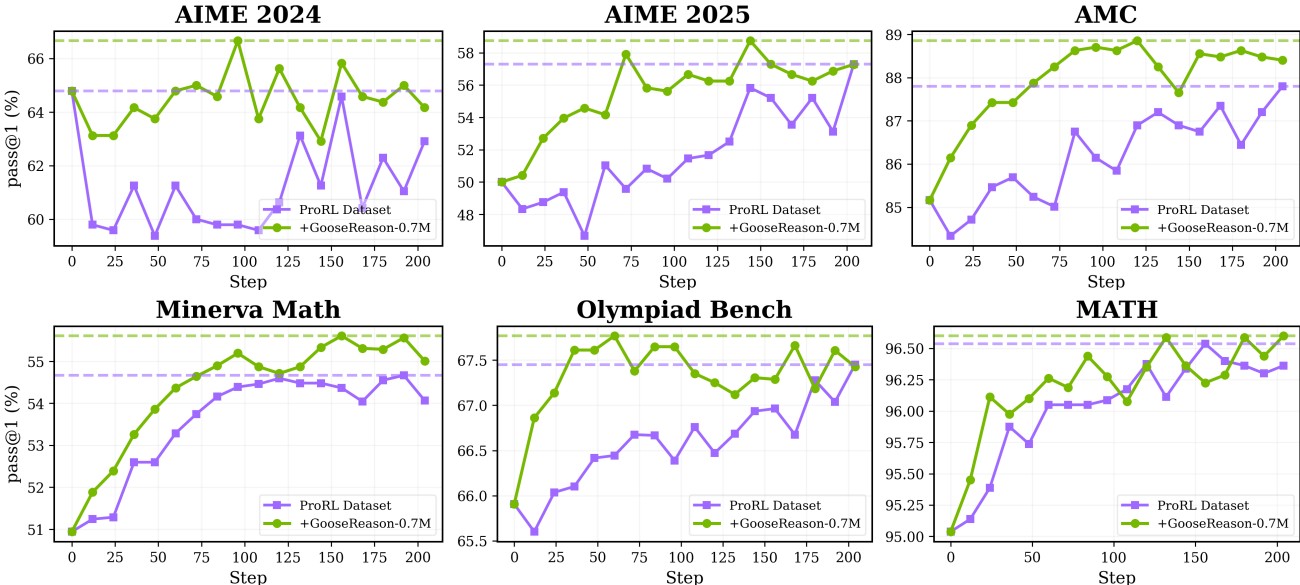

*Figure 13.* Results breakdown for Figure 6 on six math benchmarks: comparison of RL training from scratch on Qwen-4B-Instruct under a fixed compute budget with **ProRL data only** versus **joint training with GooseReason-0.7M**.

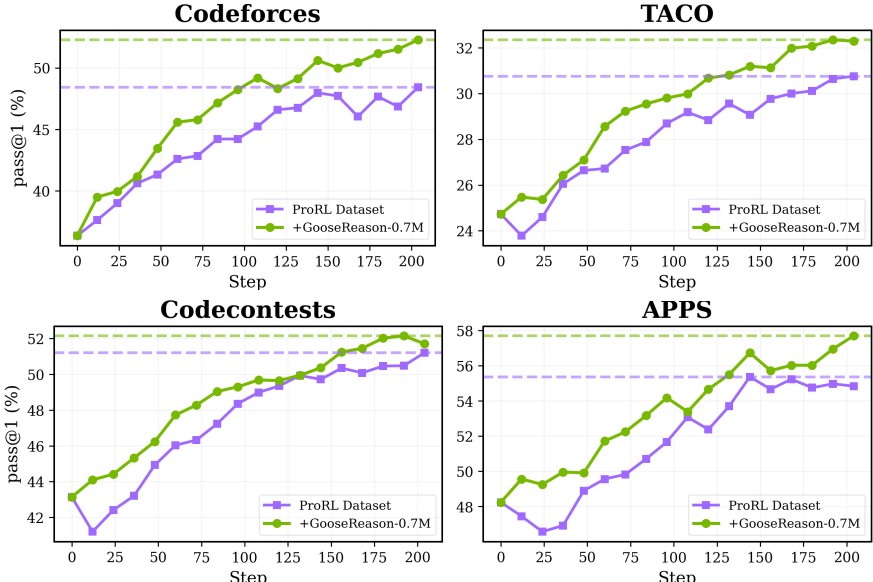

*Figure 14.* Results breakdown for Figure 6 on four coding benchmarks: comparison of RL training from scratch on Qwen-4B-Instruct under a fixed compute budget with **ProRL data only** versus **joint training with GooseReason-0.7M**.

