# OpenReview forum: "Golden Goose: A Simple Trick to Synthesize Unlimited RLVR Tasks from Unverifiable Internet Text"
_ICML.cc/2026/Conference — ICML 2026 regular_

### Official Review · Reviewer_bXD4 · 2026-03-07

**Soundness:** 3
**Presentation:** 3
**Significance:** 3
**Originality:** 4
**Overall Recommendation:** 4
**Confidence:** 4

**Summary:**

This paper introduces Golden Goose, a novel data-synthesis method for RLVR. The main idea of this paper is to convert a “fill-in-the-middle” tasks into multiple choice (fill in the [MASK]) questions, which is simple and surprised with it's effectiveness. The idea is done by a few steps: selecting a contiguous span of key reasoning steps, mask it, generate several plausible options, and then using an exact-match check on the correct option. This is then treated as the "VR" within RLVR. Empirically, the authors report improvements across 15 benchmarks in a consistent way.

**Compliance With Llm Reviewing Policy:**

Affirmed.

**Final Justification:**

While the authors addressed my concerns, there are remaining concerns which I have listed in the rebuttal response, therefore I will be maintaining my score.

**Key Questions For Authors:**

1. The method uses GPT-5. How does it perform with other (open-source) teacher models such as Qwen3 or similar? This could improve cost and scalability significantly. In this case, could you provide an ablation across teacher models, showing a quality-vs-cost curve?

2. Will the dataset be released open-source? Since the core dependency is a closed model (GPT-5) for synthesis, prompts alone may be hard to reproduce and some people may not be able to generate with the costs. Reporting costs would be helpful as well.

**Strengths And Weaknesses:**

**Strengths**

The proposed method provides a clear motivation and is both interesting and *simple*. Moreover, it is also verifiable and cheap, and mitigates instability and cost of using an LLM-as-a-judge approach during RLVR phases. The paper targets/addresses a real bottleneck in RLVR where the scaling (data) is limited by the availability of verifiable tasks and finite datasets. Empirically, the method achieves a strong performance and outperforms most of the baselines across the evaluated benchmarks. Presentation wise, the presentation is easy to follow, and the experiments and analysis align well with the stated motivation.

**Weaknesses**

1. I’m slightly concerned about relying solely on an MCQ formulation only. Providing more clarification on this would be helpful (eg, what happens if it scales beyond MCQ?). Expanding this would be really helpful

2. There are no qualitative examples in the paper. Are there cases where the pipeline labels something as incorrect even though it is actually an alternative correct answer (in other words, the distractors are not truly incorrect)? Doing this screening would be helpful in terms of analysis - understanding that sometimes it contains, but reporting it would be helpful.

3. Would there be a possibility that teacher bias and hallucinations are directly distilled into the reward target with limited detection? This may be a concern, and clarifying this further will be helpful

4. The paper appears to evaluate results only once; reporting statistical significance (e.g., standard deviation across runs) would be important.

---

> ### Author Rebuttal · Authors · 2026-03-31
>
> # Response to Reviewer bXD4
>
> Thank you for your positive review! We are glad you found the method provides "clear motivation" that is "both interesting and simple," and that it "addresses a real bottleneck in RLVR." We appreciate that you recognized the "strong performance" and found the "presentation easy to follow." We address each concern below.
>
> ## Re: Concern about relying solely on MCQ formulation
>
> Most downstream tasks we evaluate are non-MCQ (e.g., procedural math and coding problems). Injecting MCQ-style GooseReason data yields noticeable gains over ProRL data alone across these non-MCQ benchmarks (Tables 1–3), demonstrating genuine reasoning transfer rather than format-specific adaptation. Furthermore, GooseReason drives gains on tasks not directly covered by its data, such as logical puzzles in Reasoning Gym (Table 3), indicating reasoning generalization beyond both the MCQ format and the training domains.
>
> We chose the MCQ formulation as a deliberate design decision. As discussed in Section 2.3, the open-ended fill-in-the-mask alternative results in over 83% of examples with consistent zero accuracy from heavily RL-trained models due to poor instruction following, yielding no usable RL signal. The MCQ format enables simple exact-match verification without requiring an LLM-as-judge during RL, reducing both cost and instability.
>
> ## Re: No qualitative examples — are distractors truly incorrect?
>
> Please see our response to Reviewer WdkJ ("Re: Quality of synthesized supervision"), where we evaluate 2,000 MCQs with three frontier agent judges (Claude Opus 4.6, Gemini 3.1 Pro, GPT 5.3 Codex). 93.2% of tasks have all distractors truly incorrect and challenging, with 91.5% overall validity and strong inter-annotator agreement (κ=0.87). Regarding alternative correct answers: the ground-truth is the *original masked span from the source text*, so the correct answer is grounded in the source rather than being synthesized. The distractors are designed to be plausible but incorrect — our evaluation specifically checks this. We will include qualitative examples of both high-quality and edge-case tasks in the revision.
>
> ## Re: Teacher bias and hallucinations distilled into reward targets
>
> The ground-truth answer is the *original content from the source text*, not generated by the teacher — the teacher only generates distractors. This is a key design feature: the reward signal is grounded in human-authored source data, not in teacher model outputs. As shown in our response to Reviewer WdkJ ("Re: Quality of synthesized supervision"), the pipeline achieves 91.5% overall validity with strong inter-annotator agreement (κ=0.87) across three independent frontier agent judges, confirming high-quality reward targets. Furthermore, the LLM can construct valid MCQs based on source questions it can't solve. This confirms the pipeline does not require the teacher to completely comprehend or solve the source problem to produce valid reward targets.
>
> ## Re: Statistical significance
>
> All reported numbers are pass@1 averaged over 16 rollouts per question for each downstream task. The improvements are consistent across 15 diverse benchmarks spanning math, code, STEM, and logical reasoning. For Qwen3-4B-Instruct, adding GooseReason after saturation improves average math from 70.36 to 73.83 (+3.47), coding from 53.89 to 55.70 (+1.81), GPQA from 62.87 to 66.79 (+3.92), and Reasoning Gym results. This consistency across a broad set of benchmarks provides strong evidence that improvements are systematic rather than due to random variance. We will add standard deviations across multiple runs in the revision.
>
> ## Re: Performance with open-source teacher models
>
> Please see our response to Reviewer 6diA ("Re: Sensitivity to the synthesis model") for detailed ablations. We replaced GPT-5 with gpt-oss-120b and Qwen3.5-35B on the same 2,000 source questions. Overall validity is lower than GPT-5 but still relatively high (78.8% and 75.1% respectively vs. 91.5% for GPT-5). Importantly, the trend that LLMs can construct valid MCQs based on source questions they can't solve persists across all models — gpt-oss-120b and Qwen3.5-35B failed on 35.2% and 41.2% of source questions, yet synthesized valid MCQs for 68.1% and 62.4% of those.
>
> ## Re: Dataset release and cost
>
> Yes, GooseReason-0.7M will be released open-source along with the synthesis prompts (Appendix A). The cost of data synthesis is one-time — the community can directly reuse the dataset to train any model without re-running synthesis. We will include detailed cost breakdowns in the revision.

---

> > ### Author Rebuttal · Reviewer_bXD4 · 2026-04-02
> >
> > Thank you for your response. The authors have addressed most of my concerns, and I will be maintaining my score. A few remaining concerns:
> >
> > - It can only be synthesised within closed-source models. Although it's MCQ formulation, the results are not "optimal", as the open-source teacher models have a relatively large performance gap with the closed-source models, making it costly while using closed-source models.
> > - I can't really see the standard deviation across all baselines for now, so I can't comment much about whether the improvements' standard deviation overlap.
> > - I would recommend providing actual qualitative examples in the main manuscript, as it would clarify a lot and would be beneficial.
> >
> > Regardless, I'm leaning towards acceptance for this work as it's simple and effective. Good luck!

---

### Official Review · Reviewer_6diA · 2026-03-11

**Soundness:** 3
**Presentation:** 3
**Significance:** 4
**Originality:** 3
**Overall Recommendation:** 4
**Confidence:** 4

**Summary:**

This paper proposes Golden Goose, a simple pipeline for converting reasoning-rich but traditionally unverifiable text into RLVR-compatible multiple-choice tasks. The core idea is to mask a contiguous span of important reasoning steps from source text, treat the removed span as the correct answer, and generate plausible distractors so that the resulting task admits automatic verification. The authors build GooseReason-0.7M from AoPS-Instruct, rStar-Coder, and MegaScience, and GooseReason-Cyber from cybersecurity web scrapes. Experiments show that the proposed method achieves state-of-the-art results for 1.5B and 4B-Instruct models across 15 diverse benchmarks

**Compliance With Llm Reviewing Policy:**

Affirmed.

**Final Justification:**

Part of the concerns are resolved, and some claims need additional experiments to verify.

**Key Questions For Authors:**

- How sensitive is Golden Goose to the choice of the synthesis model? In particular, do the main gains persist if GPT-5 is replaced with a weaker or cheaper model for masking and distractor generation?

- How do you control for contamination or benchmark overlap when synthesizing GooseReason from large internet-scale and textbook-like corpora? This seems especially important for educational datasets and for cybersecurity web sources.

- Are there any quantitive study for the effect of choices for questions?

**Limitations:**

yes

**Strengths And Weaknesses:**

# Strengths

- The studied problem for generating large-scale synthetic data for RLVR beyond standard math and code is appealing.

- The proposed method is simple, sensible, yet reasonable.

- The empirical results are strong and practically meaningful. The paper shows two useful scenarios: scaling beyond saturation and scaling under a fixed compute budget. In both cases, GooseReason helps. For Qwen3-4B-Instruct, adding GooseReason after saturation improves the average math score from 70.36 to 73.83 and coding from 53.89 to 55.70, while also improving GPQA and reasoning-gym results.

- The proposed compute-efficient scaling result is also important, since it suggests the method is not only helpful asymptotically but also useful when RL budget is limited.

# Weaknesses
- The main weakness is that the paper relies heavily on GPT-5-based data synthesis, so it is somewhat unclear whether the major gain comes from the data or the distillation.

- Some additional studies are required to improve the soundness of the paper. For example, GooseReason is mixed into an existing ProRL-style recipe rather than evaluated in a more factorized manner. It would be helpful to better isolate the contribution of domain coverage, example freshness, example difficulty, and sheer dataset scale.

- Limited discussion about the contamination issue.

---

> ### Author Rebuttal · Authors · 2026-03-31
>
> # Response to Reviewer 6diA
>
> Thank you for your thoughtful review! We are glad you found the "studied problem for generating large-scale synthetic data for RLVR beyond standard math and code is appealing," that our method is "simple, sensible, yet reasonable," and that our "empirical results are strong and practically meaningful." We address each concern below.
>
> ## Re: Sensitivity to the synthesis model
>
> We conducted additional experiments replacing GPT-5 with gpt-oss-120b and Qwen3.5-35B as the synthesizer, using the same 2,000 source questions from AoPS. We use the same agent-as-judge evaluation as detailed in our response to Reviewer WdkJ ("Re: Quality of synthesized supervision"):
>
> | Metric                                              | GPT-5 | gpt-oss-120b | Qwen3.5-35B |
> |-----------------------------------------------------|-------|--------------|-------------|
> | MCQ format is valid                                 | 100%  | 92.3%        | 94.5%       |
> | Task requires genuine in-depth reasoning            | 95.4% | 89.2%        | 81.3%       |
> | All distractors truly incorrect and challenging     | 93.2% | 86.3%        | 84.9%       |
> | Overall valid ratio                                 | 91.5% | 78.8%        | 75.1%       |
> | Inter-annotator agreement (Cohen's κ)               | 0.87  | 0.81         | 0.82        |
> | Valid ratio for source questions LLM can't solve    | 87.2% | 68.1%        | 62.4%       |
>
> With open-source models, validity is lower than GPT-5 but still relatively high. Importantly, the trend that the LLM can construct valid MCQs based on source questions it can't solve persists across all models. gpt-oss-120b and Qwen3.5-35B failed on 35.2% and 41.2% of source questions respectively, yet synthesized valid MCQs for 68.1% and 62.4% of those — because Golden Goose asks the model to perform a relatively simple task leveraging the inherent reasoning structure in source data.
>
> The cost of data synthesis is one-time. We have open-sourced the 0.7M dataset so the community can directly reuse it without re-running synthesis.
>
> ## Re: Factorized evaluation to isolate contributions
>
> We agree this is a valuable analysis. Our paper already provides several axes of isolation:
>
> **Domain coverage.** The largest gains appear in STEM (3.48% vs. 0.13% from continued ProRL training), where GooseReason bridges the scarcity of general science RLVR data relative to the more abundant math and code domains (Section 3.1.1, Figure 5). GooseReason also drives gains on tasks not covered by its data, such as logical puzzles in Reasoning Gym (Table 3).
>
> **Example freshness.** We define an example as *effective* if it elicits both successful and failed rollouts from the policy model, yielding meaningful RL signal. As shown in Figure 3, only ~25% of the 136K ProRL samples remain effective for a saturated model — the rest become stale, with the model consistently succeeding or failing across all rollouts. GooseReason-0.7M retains ~70% effectiveness ratio, providing substantial fresh signal.
>
> **Task format and difficulty.** Section 2.3 ablates MCQ versus open-ended formulation and number of distractors (Figure 4). Open-ended results in over 83% zero-accuracy examples due to poor instruction following. With too few MCQ options (e.g., 3), the model relies on elimination rather than reasoning. With 9 options, over 70% of problems fall into a medium-difficulty regime most effective for RL.
>
> **Scaling behavior.** Figure 7 compares per-task scaling of continued RL with ProRL only versus with GooseReason-0.7M, categorized into diverge (regression vs. gain), outpace (faster gains), and align (similar trends). STEM and most math tasks diverge — where GooseReason provides the greatest benefit. Section 3.1.2 shows GooseReason enables more compute-efficient scaling under a fixed budget (Figure 6).
>
> We will add a more systematic per-domain ablation (e.g., GooseReason-Math only, GooseReason-Code only, GooseReason-STEM only) in the revision.
>
> ## Re: Limited discussion about contamination
>
> We address this in detail in our response to Reviewer UPXM ("Re: Contamination issue"). The source corpora (AoPS-Instruct, rStar-Coder, MegaScience, Primus) have undergone decontamination during their original curation, and the MCQs we synthesize are strictly grounded on them — ensuring GooseReason introduces no additional contamination for evaluation benchmarks.
>
> ## Re: Quantitative study for the effect of question choices
>
> Please see our response to Reviewer UPXM ("Re: Baselines for synthetic data"), where we compare Golden Goose against direct RLVR synthesis and direct MCQ synthesis. Golden Goose achieves notably higher validity (91.5% vs. 78.3% and 81.5%), with the gap especially pronounced on harder questions.

---

> > ### Author Rebuttal · Reviewer_6diA · 2026-04-03
> >
> > Thank you for the clarification. My concern is that the MCQ-generating model still appears substantially stronger than the base model, so this issue is not fully addressed yet. A more direct comparison might be between the model being post-trained and the model used to generate the MCQs.

---

### Official Review · Reviewer_UPXM · 2026-03-13

**Soundness:** 2
**Presentation:** 3
**Significance:** 1
**Originality:** 2
**Overall Recommendation:** 3
**Confidence:** 5

**Summary:**

The paper pose a method synthesize RLVR tasks from raw and unverifiable (pretraining) text.
Given a text, the method prompts it with an LLM to synthesize distractor and masking out key details, and produce QA pairs with masked content as ground truth.
The paper scale the data and synthesize them with GPT-5, and train the 4B-7B models with RL and show positive improvements across tasks.

**Compliance With Llm Reviewing Policy:**

Affirmed.

**Final Justification:**

The authors' rebuttal resolve some but not all concerns.

**Key Questions For Authors:**

NA

**Strengths And Weaknesses:**

* Strength
- The method is simple, easy to scale and implement.
- The method achieve good performance through scaling RL and data size with raw data.

* Weaknesses
- the paper failed to mention contamination issue. These raw text are almost certainly pre-trained by the LLM, leading it to already know or memorize the answer.
- The method did not have a clean-up or filtering phase, which is important, because the LLM may fail to comply with the instructions, produce hallucinations, incorrect data. This would produce a lot of noisy data with invalid labels.
- Gains are marginal given the huge data size increase. for example, with 2-3% improvement on math
- The paper use GPT-5 as the data synthesis model, and train the data with tiny models. This defeats the potential of whether this method could be appllicable for frontier model, which I doubt it would.
- There are many different ways of doing this, and many baselines that involve producing synthetic data for RL, which the paper didn't mention or compare with.

---

> ### Author Rebuttal · Authors · 2026-03-31
>
> # Response to Reviewer UPXM
>
> Thank you for your review. We address each concern in detail below.
>
> ## Re: Contamination issue
>
> **Overlap with evaluation benchmarks.** The source corpora (AoPS-Instruct, rStar-Coder, MegaScience, Primus) have already undergone filtering and decontamination during their original curation. The MCQs we synthesize are strictly grounded on these corpora, ensuring GooseReason introduces no additional contamination.
>
> **Source text seen during pre-training.** Even if the policy model (e.g. Qwen3-4B) encountered source passages during pre-training, Golden Goose tasks remain challenging. As shown in Figure 4, over 80% of tasks from GooseReason-0.7M have accuracy below 0.85 across 16 rollouts from the policy model, yielding meaningful learning signal for RL.
>
> ## Re: No clean-up or filtering phase
>
> Please see our response to Reviewer WdkJ ("Re: Quality of synthesized supervision"), where we evaluate 2,000 MCQs with three frontier agent judges. The pipeline achieves 91.5% overall validity with strong agreement (κ=0.87) without heavy filtering. The LLM can even construct valid MCQs based on source questions it can't solve (87.2% validity on the 26.7% GPT-5 fails).
>
> ## Re: Gains are marginal given data size
>
> These gains are achieved *after saturation*. The baselines have been trained to saturate using ProRL with over 140K human-curated RLVR data, where further training yields no improvement or degrades performance. Any gain beyond saturation is significant as genuinely new learning signal. For Qwen3-4B-Instruct, adding GooseReason after saturation improves average math from 70.36 to 73.83 (+3.47), coding from 53.89 to 55.70 (+1.81), and also improves GPQA and Reasoning Gym — consistent improvements spanning 15 diverse benchmarks. Importantly, most evaluation tasks are non-MCQ (e.g., procedural math/coding problems). Injecting MCQ-style GooseReason data yields noticeable gains over ProRL data alone, demonstrating generalizability and transferability.
>
> ## Re: GPT-5 and training tiny models
>
> Please see our response to Reviewer 6diA ("Re: Sensitivity to the synthesis model") for ablations with gpt-oss-120b and Qwen3.5-35B. GPT-5 is used once to construct the dataset; we have open-sourced the data for the community to reuse directly. The focus on 1.5B and 4B models is due to compute constraints during RL. The data itself is model-agnostic and applicable to larger models.
>
> ## Re: Baselines for synthetic data
>
> We compare Golden Goose against two alternatives using GPT-5 based on the same 2,000 source questions: (1) **Direct RLVR synthesis**: synthesize a question with a math-parser-verifiable answer. (2) **Direct MCQ synthesis**: directly synthesize an MCQ.
>
> **All 2,000 samples:**
>
> | Metric                                              | Golden Goose | Direct RLVR | Direct MCQ |
> |-----------------------------------------------------|--------------|-------------|------------|
> | Question format is valid                            | 100%         | 87.1%       | 98.7%      |
> | Task requires genuine in-depth reasoning            | 95.4%        | 82.1%       | 85.9%      |
> | All distractors truly incorrect and challenging     | 93.2%        | N/A         | 83.2%      |
> | Overall valid ratio                                 | 91.5%        | 78.3%       | 81.5%      |
>
> **Source questions GPT-5 cannot solve (26.7%):**
>
> | Metric                                              | Golden Goose | Direct RLVR | Direct MCQ |
> |-----------------------------------------------------|--------------|-------------|------------|
> | Question format is valid                            | 100%         | 75.5%       | 97.6%      |
> | Task requires genuine in-depth reasoning            | 91.2%        | 79.1%       | 73.1%      |
> | All distractors truly incorrect and challenging     | 90.9%        | N/A         | 78.3%      |
> | Overall valid ratio                                 | 87.2%        | 73.3%       | 70.5%      |
>
> Both alternatives show notably lower validity. Directly synthesizing tasks has higher cognitive load than Golden Goose's simpler masking key reasoning steps + generating distractors. The gap is especially pronounced on questions the model cannot solve itself. Two noticeable failure modes: 12.9% of direct RLVR answers are unparsable by the math verifier, and both alternatives tend to produce simpler questions even from complex sources.
>
> ## Re: Novelty and significance
>
> Golden Goose addresses data saturation and the difficulty of constructing large-scale RLVR data. It provides a scalable path for transforming reasoning-rich unverifiable corpora previously excluded from RLVR into verifiable tasks that fuel continued RL scaling. Though simple, it revives saturated models with robust gains across math, code, and STEM, achieving new SOTA for 1.5B and 4B-Instruct across 15 benchmarks. The simplicity makes it adaptable to new domains with only a prompt change, including cybersecurity where no prior RLVR data exists.

---

> > ### Author Rebuttal · Reviewer_UPXM · 2026-04-02
> >
> > Thanks for the rebuttal. Many of my concerns remain. But I bumped the score

---

### Official Review · Reviewer_WdkJ · 2026-03-13

**Soundness:** 3
**Presentation:** 3
**Significance:** 3
**Originality:** 2
**Overall Recommendation:** 4
**Confidence:** 4

**Summary:**

This paper addresses a key bottleneck in reinforcement learning with verifiable rewards (RLVR): the scarcity of large-scale verifiable training data. The authors propose Golden Goose, a simple pipeline that converts reasoning-rich but unverifiable internet text into verifiable multiple-choice RLVR tasks via a fill-in-the-middle reformulation. Given a source passage, an LLM identifies a contiguous span of important reasoning steps, masks it, and then generates multiple plausible but incorrect distractors, turning the original text into a multiple-choice question with an automatically checkable answer. Using this pipeline, the paper constructs GooseReason-0.7M, a dataset of over 0.7M tasks spanning math, code, and STEM, and also builds GooseReason-Cyber from cybersecurity web scrapes. Empirically, the paper shows that these synthetic datasets can revive models that have saturated on existing RLVR data, improve compute efficiency under fixed RL budgets, and achieve strong benchmark results, including new state-of-the-art performance for 4B instruct models and competitive results in cybersecurity.

**Compliance With Llm Reviewing Policy:**

Affirmed.

**Final Justification:**

I appreciate the authors thorough response, and I will maintain my score.

**Key Questions For Authors:**

1. The method depends critically on the quality of the masked spans and distractors produced by GPT-5. Can the authors provide a more direct quality analysis, such as human evaluation, ambiguity rate, or error taxonomy for the synthesized tasks? A strong answer would increase my confidence in the data contribution.
2. How much of the gain comes from the specific Golden Goose formulation, as opposed to simply adding more synthetic post-training data from the same source corpora? Comparisons to simpler synthetic baselines would help clarify the contribution.
3. The title emphasizes “unlimited” RLVR tasks. In practice, what are the main bottlenecks: source availability, synthesis cost, filtering cost, or quality degradation at scale? A more careful discussion would strengthen the paper’s claims.
4. The paper argues that the method improves general reasoning beyond the MCQ format. Can the authors provide more evidence that the gains are not primarily due to training on a multiple-choice task family, but instead reflect broader transferable reasoning improvements?

**Limitations:**

Partially. The paper includes an impact statement and briefly acknowledges dual-use concerns in cybersecurity, as well as the possibility that source-text biases or toxic content may be inherited. However, I think the paper should discuss limitations more explicitly, especially regarding data quality control, reliance on a powerful teacher model for synthesis, and the risk of amplifying errors or biases from noisy web corpora.

**Strengths And Weaknesses:**

# Strengths
+ Addresses an important bottleneck. The paper focuses on a highly relevant issue for current RLVR research: performance saturation caused by limited effective verifiable data. This is a real and timely problem for scaling reasoning models.
+ Simple and intuitive core idea. The central trick—recasting unverifiable reasoning-rich text into a verifiable multiple-choice fill-in-the-middle task—is easy to understand and practically appealing. It is also flexible enough to be applied to multiple source domains.
+ Strong data-centric contribution. The paper does not propose yet another RL optimization variant, but instead expands the supply of RLVR data. This makes the work complementary to prior scaling efforts based on algorithms, rollouts, or handcrafted environments.
+ Large-scale dataset construction. GooseReason-0.7M is substantial in size and covers diverse domains including mathematics, programming, and science. The cybersecurity extension is also a nice demonstration that the method can be deployed beyond standard reasoning benchmarks.
+ Empirical results are convincing overall. The paper reports improvements in continued RL after saturation, under fixed compute budgets, and in a specialized cybersecurity setting. The comparisons against continued training on the original ProRL data are especially useful and support the main claim that new RLVR data can revive stalled training.
+ Some design choices are empirically motivated. The ablation on multiple-choice versus open-ended formulations and the analysis of the number of distractors are useful and help justify the final design.

# Weaknesses
+ The novelty is somewhat limited. While the paper is practically useful, the underlying idea is fairly straightforward: use a strong LLM to transform existing text into multiple-choice cloze-style questions, then use them as RLVR tasks. This is a nice trick, but conceptually it feels more like a clever data engineering pipeline than a major methodological advance.
+ Quality of synthesized supervision is not deeply analyzed. The paper relies heavily on GPT-5 to identify masked reasoning spans and generate distractors, but there is limited analysis of synthesis quality, ambiguity, distractor hardness, or noise rates. Since the whole contribution depends on data quality, I would like more direct validation here.
+ The claimed “unlimited” framing feels overstated. In practice, the usefulness of the data still depends on source quality, prompt quality, synthesis cost, and task filtering. The method may be scalable, but “unlimited RLVR tasks” is rhetorically stronger than what is actually demonstrated.
+ It is unclear how much the gains come from format conversion versus genuine reasoning transfer. The paper argues that improvements transfer to non-MCQ benchmarks, which is encouraging, but it remains hard to disentangle whether the gains come from better reasoning skills, better calibration to multiple-choice-style supervision, or simply more diverse post-training data.
+ Baselines could be stronger in the data-construction dimension. The paper compares against continued training on ProRL and against RLVE in some settings, but I would have liked to see stronger baselines using simpler synthetic data generation approaches, such as standard QA reformulation, self-generated MCQs without reasoning-span masking, or other lightweight data augmentation pipelines. This would help isolate the real added value of Golden Goose.

---

> ### Author Rebuttal · Authors · 2026-03-31
>
> # Response to Reviewer WdkJ
>
> Thank you for your constructive review! We are pleased you recognized our "simple and intuitive core idea," "strong data-centric contribution," and found our "empirical results convincing overall." We address each concern below.
>
> ## Re: Quality of synthesized supervision is not deeply analyzed
>
> We evaluated 2,000 randomly sampled MCQs from GooseReason-0.7M (math domain, AoPS source). Given the Olympiad-level difficulty, we use an agent-as-judge approach with three frontier models (Claude Opus 4.6, Gemini 3.1 Pro, GPT 5.3 Codex), each equipped with web browser, search, and code execution. Each MCQ is evaluated on three binary criteria: (1) **Format validity**: any format issues such as too few or duplicated distractors, or correct answer significantly longer/shorter than distractors. (2) **Reasoning effort**: whether deriving the correct answer requires non-trivial, in-depth reasoning. (3) **Distractor validity**: whether each distractor is truly incorrect and stylistically similar enough to the correct answer to be challenging. A question is valid only if all criteria are met and all three judges agree.
>
> | Metric                                              | Rate  |
> |-----------------------------------------------------|-------|
> | MCQ format is valid                                 | 100%  |
> | Task requires genuine in-depth reasoning            | 95.4% |
> | All distractors are truly incorrect and challenging | 93.2% |
> | Overall valid ratio                                 | 91.5% |
> | Inter-annotator agreement (Cohen's κ)               | 0.87  |
>
> High quality is achieved without heavy filtering, with strong agreement (κ=0.87). We attribute this to the simplicity of the synthesis task: identify reasoning steps to mask and generate distractors — substantially easier than *solving* the source problem.
>
> **The LLM can construct valid MCQs based on source questions it can't solve.** GPT-5 failed to solve 26.7% of the 2,000 source questions, yet synthesized MCQs for those questions remain valid at 87.2%, comparable to the overall 91.5%. This highlights that constructing a well-formed MCQ is fundamentally easier than solving the original problem.
>
> ## Re: Baselines could be stronger
>
> Please see our response to Reviewer UPXM ("Re: Baselines for synthetic data") for full details. In summary, we compare Golden Goose against (1) direct RLVR synthesis (synthesize a math-parser-verifiable question) and (2) direct MCQ synthesis, using GPT-5 on the same source data. Golden Goose achieves notably higher validity (91.5% vs. 78.3% and 81.5%). The gap is especially pronounced on harder questions that the model cannot solve (87.2% vs. 73.3% and 70.5%), as directly synthesizing RLVR tasks imposes a higher cognitive load than Golden Goose's approach of masking key reasoning steps from source data.
>
> ## Re: The novelty is somewhat limited
>
> Golden Goose addresses an important bottleneck: data saturation on existing RLVR data and difficulty to automatically construct large-scale RLVR data. It provides a scalable path for transforming abundant, reasoning-rich, yet unverifiable corpora — previously excluded from RLVR data curation — into high-quality verifiable tasks that fuel continued RL scaling. The practical impact is a 13× increase in effective RL training examples over the ProRL data blend (Figure 3), reviving saturated models with the largest gains in STEM (3.48% vs. 0.13%), precisely where existing RLVR data is scarcest. The simplicity also makes it scalable and adaptable to new domains with only a prompt change (Appendix A).
>
> ## Re: Format conversion versus genuine reasoning transfer
>
> Please see our response to Reviewer bXD4 ("Re: MCQ formulation"). We showed in paper that gains transfer to non-MCQ benchmarks and uncovered domains like Reasoning Gym. Most evaluation tasks are procedural math and coding problems, yet injecting MCQ-style GooseReason data yields consistent improvements over ProRL data alone.
>
> ## Re: "Unlimited" framing
>
> "Unlimited" refers to the source — reasoning-rich internet text is effectively unbounded — though in practice, as the reviewer notes, usefulness depends on source quality, synthesis cost, and filtering. That said, we observe consistently high quality at scale with no degradation across our 0.7M tasks spanning four domains, including noisy FineWeb scrapes. We will include a detailed discussion of practical scaling considerations in the revision.
>
> ## Re: Dependency on GPT-5
>
> Please see our response to Reviewer 6diA ("Re: Sensitivity to the synthesis model"). In summary, replacing GPT-5 with gpt-oss-120b and Qwen3.5-35B yields lower but still relatively high validity (78.8% and 75.1%). The trend that LLMs can construct valid MCQs based on source questions they can't solve persists across all models.

---

> > ### Author Rebuttal · Reviewer_WdkJ · 2026-04-04
> >
> > I appreciate the authors thorough response, and I will maintain my score.

---

### Decision · Program_Chairs · 2026-04-30

**Decision:**

Accept (regular)

**Comment:**

The paper provides a simple method to generate training data for RLVR methods. Using masking, it converts existing data to a MCQ question. While the method is straightforward, the experiments show the value of this technique. It can be easily applied to training runs for SLMs, especially in domains such as cybersecurity that may not have the training data with verifiable rewards. Ablations on design choices and the contribution of a 0.7M dataset is also welcome.

Reviewers raised a valid concern that this method cannot be used to train frontier models, since the data generation LLM is typically much stronger than the target model. I suggest the authors to clarify this in their manuscript. Overall, there is still significant utility of this work for training small models.